# Antibiotic Resistance in Minimally Human-Impacted Environments

**DOI:** 10.3390/ijerph17113939

**Published:** 2020-06-02

**Authors:** Laura C. Scott, Nicholas Lee, Tiong Gim Aw

**Affiliations:** Department of Environmental Health Sciences, School of Public Health and Tropical Medicine, Tulane University, New Orleans, LA 70112, USA; lscott9@tulane.edu (L.C.S.); nlee10@tulane.edu (N.L.)

**Keywords:** antibiotic resistance, environment, anthropogenic activity, methodology

## Abstract

Antibiotic resistant bacteria (ARB) have become contaminants of concern in environmental systems. Studies investigating environmental ARB have primarily focused on environments that are greatly impacted by anthropogenic activity. Background concentrations of ARB in natural environments is not well understood. This review summarizes the current literature on the monitoring of ARB and antibiotic resistance genes (ARGs) in environments less impacted by human activity. Both ARB and ARGs have been detected on the Antarctic continent, on isolated glaciers, and in remote alpine environments. The methods for detecting and quantifying ARB and ARGs from the environment are not standardized and warrant optimization. Further research should be focused on the detection and quantification of ARB and ARGs along human gradients to better characterize the factors leading to their dissemination in remote environments.

## 1. Introduction

The presence of antibiotic resistant bacteria (ARB) in the environment are commonly associated with hospitals and their frequent use and disposal of antibiotics [1]. ARB are a persistent contaminant in environments in close contact with humans [2]. The antibiotic resistant genes (ARGs) within ARB can proliferate through environments due to persistent human activity and the perpetuation of the genes by horizontal gene transfer and vertical inheritance [3]. These ARGs can establish reservoirs in the environment particularly near human activities such as wastewater treatment plants, agricultural operations, and hospitals [4]. The common use of antibiotics and the subsequent selection for ARB in the human microbiome creates a persistent population of ARB in areas of regular human contact via human waste [1].

Numerous studies have been focused on the presence of ARB or ARGs in environments with repeated or close human contact, such as wastewater effluent and animal agriculture settings [2,4,5]. These environments have been extensively studied in regard to antibiotic resistance—at least eight literature reviews have been published summarizing ARB and ARGs in environmental compartments associated with high human impact [4,5,6,7,8,9,10,11]. These studies contribute to the characterization of ARGs and ARB in areas of greater human impact, however there are fewer studies investigating the presence of ARB in pristine or minimally human-impacted environments [8]. Investigating these environments for the presence of ARB may help to determine the extent to which the larger ecosystem has been impacted by human activities. Furthermore, the study of minimally human-impacted environments may help elucidate the extent of contamination of ARB and ARGs from anthropogenic sources. A research need has previously been identified to determine background or intrinsic levels of antibiotic resistance in soil in order to quantify contamination by resistant bacteria in soil environments [12]. The issue of environmental antibiotic resistance can be addressed within the One Health paradigm. In order to estimate potential human health risks, the effects of environmental ARGs and ARB on wildlife and the surrounding ecological systems must be understood and quantified. The quantification of ARB and ARGs in minimally human-impacted environments provides perspective on background levels of these contaminants, scope of the extensiveness of contamination, and insight into the dissemination mechanisms of these contaminants into and within the environment.

Bacteria and other microorganisms were using antibiotics as a competitive mechanism long before the human development of antibiotic drugs [13]. Remnants of this ancient biological warfare mechanism can be detected in pristine environments [13,14,15]. These ARGs and ARB evolving in isolation from anthropogenic input can be considered the “background” antibiotic resistance present in any given environmental bacterial community. It can therefore be expected to detect ARGs or ARB in most environmental compartments, making it precarious to discern “background” from “contamination”. Soil environments have previously demonstrated increased abundances of ARGs compared to water, likely due to this background [14]. It can be hypothesized that in areas of minimal human impact, these background levels of antibiotic resistance would be greater in soil than water.

The focus of environmental ARGs and ARB as a One Health issue has been a research priority since 2004 [16]. Molecular biology methods, particularly “omics” technologies are booming, but have not been standardized for this subfield. Much of the terminology is nonstandardized and adds to the difficulty of determining consensus. The objectives of this review were: (1) amalgamate and describe studies investigating ARB and ARGs in minimally human-impacted environments or along human gradients, (2) illustrate global trends of environmental ARB and ARGs, (3) review current analytical methods within these studies, and (4) provide recommendations for future studies in this field. For the purposes of this review, minimally human-impacted environments have been defined as nonresidential, nonagricultural, highly inaccessible, or government protected sites. These environments include water, soil, and ice in polar regions, remote glaciers, lakes, and alpine sites, and deep ocean sediments. The search engines PubMed and Google Scholar were queried using the terms “antibiotic resistance”, “environment”, and “pristine”. For the purposes of this review, studies investigating ARGs or ARB in polar regions, highly remote sites, governmentally protected environments, or self-proclaimed to be removed from anthropogenic impact were included. Of the identified publications, 23 were reviewed and are summarized in Table 1.

## 2. Detection of ARB and ARGs in Polar Regions

Polar regions might be considered some of the most minimally human-impacted environments due to their harsh conditions and inaccessibility. The presence of scientific stations on the Antarctic continent creates a gradient of human impact that has been used to study antibiotic resistant bacteria and genes. Recession of glaciers in the high Arctic has also provided insight to potentially ancient mechanisms of antibiotic resistance.

### 2.1. Antarctica

The proximity to Antarctic scientific stations was positively correlated with the detection of ARGs or mobile genetic elements (MGEs) in two studies [17,21]. Bacterial isolates from Antarctic sewage water had more resistant bacteria than faunal fecal isolates on McMurdo Sound [18]. Metagenomic analysis has demonstrated that efflux pumps, bypass mechanisms, target modification, and target inactivation are the most commonly detected ARG types from soils on King George Island [22]. Samples from far inland Antarctica and sites beneath the Mackay Glacier suggested interspecies competition and vertical inheritance contributed to the presence of ARGs, suggesting ancient mechanisms of antibiotic resistance [15,20].

### 2.2. Arctic

In Canada’s high Arctic, 84% of *coliform* isolates from glacial ice and water were resistant to cefazolin, 71% resistant to cefamandole, and 65% resistant to ampicillin [23]. However, no isolates were resistant to streptomycin, gentamicin, chloramphenicol, or ciprofloxacin [23]. Principal component analysis of ARG variation from Kongsfjorden soil samples revealed that 67% was due to lithological factors or nutrients, and that multidrug efflux pumps accounted for 30% of the ARG types detected [25]. Additionally, MGEs were significantly associated with ARG abundances suggesting horizontal gene transfer [25]. Another study in Kongsfjorden assessed antibiotic resistance of heterotrophic bacterial isolates from water and sediment, and determined water isolates were most resistant to beta-lactams, while sediment isolates had more variable resistance profiles [26]. ARGs have also been detected in surface sediments in the Bering Sea and from the Polar Research Institute [14]. Sulfonamide genes (*sul1*, *sul2*, and *sul3*) were most prevalent, and 87% of targeted ARGs were detected in the Arctic and sub-Arctic samples [14].

## 3. Detection of ARB and ARGs in Minimally Impacted Nonpolar Regions

Other environments in nonpolar regions, such as glaciers, caves, and deep oceans, could be considered as minimally impacted due to their inaccessibility or governmental protection [33,34,36]. Both water and soil have been studied for ARB and ARGs in these types of “pristine” environments.

### 3.1. Water Environments

Several studies have examined water for ARB and ARGs in nonpolar minimally human-impacted environments, including freshwater and saltwater sources [27,31]. Isolates from a river system within Serra de Cipo National Park in Brazil were tested for resistance against 11 antibiotics [27]. A total of 93% of isolates were resistant to at least one antibiotic, 61% had multiple resistance overall, and 77% of isolates taken during the rainy season had multiple resistances [27]. Interestingly, biotic factors including pH, water temperature, and dissolved organic carbons were not associated with antibiotic resistance patterns, contrary to findings from an urban river in India [27,39]. Twenty-one Swiss Lakes were sampled at their deepest point and in the upper 5 m and assessed for six ARGs [28]. Sulfonamide genes were most common, followed by tetracycline genes, and the fluoroquinolone resistance gene, *qnrA*, was not detected in any of the samples [28]. Regression analysis indicated that variation in the *sul1* gene was best explained by the presence of wastewater treatment plants and hospital effluent, but there was no significant difference in sulfonamide genes between human-impacted and non-human-impacted lakes [28]. A study of remote, high elevation lakes in Argentina found that all isolated bacteria were resistant to at least 8 of 11 tested antibiotics, and every isolate was resistant to ampicillin and macrolides used [29]. Despite the lack of mechanistic evidence, the study correlates spore forming ability with increased ARB abundance, consistent with previous investigations [7,29]. Similarly, an investigation of five hot springs in the Limpopo Province of South Africa found that 40 isolates from water samples were resistant to both antibiotics and heavy metals, 15 of which had multiple antibiotic resistances [30]. In Lechuguila Cave, a permit access-only cave within the United States National Park Service, 70% of the 93 of Gram-positive bacterial isolates were resistant to three or four antibiotic classes, and 65% of Gram-negative isolates were resistant to three or four antibiotic classes [33]. Fifty *Vibrio* isolates were obtained from polluted and pristine reserves in Hong Kong, and all were resistant to ampicillin, carbenicillin, cephalothin, clindamycin, colistin, erythromycin, fusidic acid, methicillin, nitrofurantoin, and penicillin [31]. In addition, all samples from a protected reef system in Georgia, United States tested positive for either a tetracycline or integrase gene, with *tetM*, *tetQ*, and *int1* being the most commonly detected genes [32].

### 3.2. Soil/Sediment Environments

A study examining sediment from the Poudre River and South Platte River, which have headwaters in Rocky Mountain National Park and runs through the Roosevelt National Forest, found ARGs present in all samples. However, ARGs were more frequently detected in downstream sites than near the headwaters [34]. Similarly, soil samples from a variety of land uses and anthropogenic activity were taken from Ningbo, Zhejian Province in China and compared to determine if level of anthropogenic activity or seasonality affected ARG detection or abundance [35]. The study demonstrated that the number of ARGs detected in farmland soil was significantly higher than that in forest soil in Spring, Summer, and Winter months [35]. A study analyzing sediments taken from a high human impact estuary of South China and deep ocean sediments within the South China Sea found that relative ARG abundance trended with concentrations of antibiotics found in the same samples [36]. In remote areas of Tibet, ARGs were more abundant in lake sediments than soil or animal waste, and only 2% of identified ARGs were carried by plasmids, demonstrating the propensity for sediments to be an environmental reservoir of ARGs [37]. An extensive study of 51 glacier sites from across the globe (Central Asia, North America, South America, Greenland, Himalayas, Africa, and Antarctica) found that glaciers near more populated areas had the highest relative ARG abundances [38].

## 4. Analytical Methods Used for the Detection and Quantification of Environmental ARB and ARGs

### 4.1. Determining Antibiotic Susceptibility of Environmental Bacteria

There are currently no standardized methods for antibiotic susceptibility testing of environmental bacteria. In general, analytical methods used for investigating environmental ARB consist of obtaining environmental media, isolating bacteria from the media sample, and testing isolates for resistance against selected antibiotics. While this method is consistent with clinical isolate testing, it is not necessarily the most informative or efficient for environmental monitoring. Assessing resistance of single isolates provides a narrow snapshot of the phenotypic resistance patterns in the microbial community. Rather than assessing antibiotic resistance of isolates at random, growing bacteria on generalized media in the presence of selected antibiotics and without antibiotics would allow for assessing relative resistance within the bacterial community of that sample, identifying which members of the community are phenotypically resistant. Assessing isolates from this type of analysis for multidrug resistance offers a broad perspective on resistance within the community as well as resistance within individual isolates. However, the utilization of generalized media to determine antibiotic susceptibility among a larger variety of bacteria taxa will result in a competitive bias towards fast-growing groups.

The Clinical Laboratory Standards Institute currently has standards for antibiotic susceptibility testing and identification of clinical isolates, the disc diffusion method, determination of minimum inhibitory concentrations (MIC), and automated susceptibility instruments [40]. These methods could be easily transferred to investigations of environmental ARB if an indicator organism, such as *Escherichia coli* or *Staphylococcus aureus*, is used. The integration of indicator organisms would accelerate widespread monitoring of ARB in environmental compartments at the expense of understanding the distribution of ARB throughout the microbial community. However, human-associated, well-described indicator species will also be extremely rare or absent in most environments with minimal human activity and this could introduce sampling bias. If these methods are applied to unknown communities of bacteria, inherent confounding will exist due to Gram status and phylogenic intrinsic antibiotic resistance. Therefore, whether the MIC method or a single concentration of antibiotic is used when investigating antibiotic susceptibility on a microbial population, a bias will occur.

### 4.2. Flow Cytometry as a High-Throughput Culture Alternative

In addition to traditional culture methods, flow cytometry (FCM) has recently been utilized as a tool for quantifying environmental bacteria [41,42,43,44]. Flow cytometers are instruments utilizing lasers to discriminate particles by size and shape, and can incorporate fluorescent tagging [45]. The combination of fluorescent cell permeable and impermeable nucleic acid probes and light scattering properties of bacterial particles allows for the viability assessment of bacterial communities Calcein violet-acetoxymethyl ester (CV-AM) has been tested to measure the susceptibility of *Mycobacterium tuberculosis* to a variety of bacteriostatic antibiotics and was found to be reliable [46]. Flow cytometry with a propidium iodide-SYTO costaining technique has been used for rapid detection of urinary tract infections, and was found to have a sensitivity of 100% and specificity of 78.4% [47,48]. This dual staining technique with propidium iodide, a viability assessment fluorophore, offers the potential for assessing environmental bacteria for resistance to bactericidal antibiotics. Flow cytometry has been used to measure bacterial communities in complex environmental samples such as plant roots, wastewater sludge, treated wastewater effluent, and agricultural soils [42,43,44]. The development of a high throughput method comparable to standard culture method allows for more thorough examination of antibiotic resistance in the environment speciation of resistant or susceptible bacteria might not be possible with current flow cytometry technology. While much work has been done to evaluate flow cytometry for measuring bacterial viability and vitality, no work has been done to demonstrate the efficacy of FCM for the examination of antibiotic susceptibility of complex microbial populations in environmental samples.

### 4.3. Molecular Techniques for Analysis of ARGs

Molecular techniques such as quantitative polymerase chain reaction (qPCR) are currently the preferred method for the identification and quantification of ARGs [14,28]. Few studies integrated metagenomic analyses, an untargeted genomic analysis capable of characterizing both the scope of antibiotic resistance genes present in a sample and the taxonomic groups present in analyzed samples [15,22,25,36,37,38]. Metagenomics is a potentially useful tool for the investigation of ARB and ARGs in low impact environments, particularly in an attempt to compare how microbiomes are changing under anthropogenic selection pressures. The reviewed literature demonstrates that the type of ARGs present in environmental samples vary widely across environments and environmental compartments, necessitating a screening analysis prior to targeted analyses. The utilization of untargeted analyses available via shotgun DNA sequencing and bioinformatics could be used to dictate subsequent targeted (qPCR) ARG analysis. The combination of genomic and culture-based methods for examining ARB and ARGs in natural environments informs on bacterial viability, pathogenicity, and quantity, which are imperative for assessing health risk.

The study of antibiotic resistance in environmental compartments is in need of methodology standardization. No single method for investigating environmental ARB and ARGs is ideal (Table 2). When designing environmental ARB and ARG studies, careful consideration should be given to the overall objective. Research seeking to inform on human or animal risk might choose methods more similar to clinical methods such as incorporating a pathogen indicator, MIC determination, and specific antibiotic gene targets. However, studies aimed at characterizing environments, investigating dissemination of ARB or ARG within the environment, or determining factors associated with the presence and abundance of ARB/ARGs should opt for broad spectrum methods such as FCM culture analysis and metagenomic profiling followed by targeted genomic analysis.

## 5. Conclusions

This review demonstrates that ARB and ARGs can be detected in a multitude of environments and environmental media across the globe. While the argument has previously been made that no place on Earth can be considered “pristine”, multiple studies have demonstrated the associations between human activities with the presence and abundance of ARB/ARGs [17,21,34,36,38]. Additionally, while few sampling sites have been found to be free of ARGs, ARB are not ubiquitously detected in minimally impacted sites [17,19]. This demonstrates a need for more research to be conducted along human activity gradients to better understand the distinct role of human activity on the presence of ARB and ARGs in environmental compartments.

Several studies also illustrated that evidence of ancient antibiotic resistance, due to ecological competition, can be found and differentiated from plasmid genes in minimally human-impacted environments [15,20,37]. The ability to differentiate between these mechanisms of resistance is an important contribution to the assessment of human impact regarding antibiotic resistance. ARGs commonly arising in association with human impact could potentially be used as markers of anthropogenic activity [14].

Coresistance between heavy metals and antibiotics has been demonstrated in minimally human impacted-environments [29,30]. Bacterial exposure to cytotoxic substances, such as heavy metals, is associated with cross-resistance mechanisms such as horizontal gene transfer, resulting in nonspecific efflux pumps that confer resistance to a multitude of chemicals, including antibiotics [49]. Other factors, including media temperature, pH, and other chemical/physical parameters have been investigated as determinants of the presence and abundance of ARB/ARGs [27,28]. However, the factors selected are not uniform, and the results remain unclear. In complex environments, it is likely that many biotic and abiotic factors contribute to the presence of ARB/ARGs, and this should be further investigated.

Based upon studies in this review that reported sample size, the number of environmental media samples taken ranged from 3 to 105, averaged 26 samples, with 8 samples being the most common. Increased sample sizes would likely elucidate statistical associations, contributing to a better understanding of how anthropogenic activity affects the presence and abundance of ARB in minimally impacted environments. Additionally, increased sampling along human gradients will reveal the extent of ARB contamination in soil and water environments as well as patterns associated with their dissemination. However, the inclusion of more samples may introduce the possibility of varying microenvironments and the bias associated with the differing bacterial communities within them. Nevertheless, further investigation of these changes and their effect on the bacterial community while holding constant anthropogenic impact offers the opportunity to characterize these potential drivers of resistance.

Finally, the definition of “minimally human impacted” or “pristine” is subjective, undefined, and inconsistent among studies. Studies in far polar regions, deep oceans, and hot springs demonstrate isolated environments with consistently low human impact over time. It is unlikely that the degree of anthropogenic impact is equal at sites nearer to human civilization. Future studies should consider incorporating a metric of human activity in an attempt to standardize and quantify human activity, in the absence of a definition of “pristine”.

Antibiotic resistance is an imminent threat, and examination of environments, including those with minimal human impact, is essential. The mechanism of ARB and ARG dissemination into and within the environment is understudied and not well understood. Minimally human-impacted environments represent ideal candidates for tackling these pivotal research questions.

## Figures and Tables

**Table 1 ijerph-17-03939-t001:** Summary of studies investigating the presence of antibiotic resistant bacteria (ARB) and antibiotic resistance genes (ARGs) in minimally human-impacted environments.

Sampling Location	Sample Media	ARB/ARGs/Antibiotics	Key Findings	Reference
Antarctica	Soil	ARGs	19% of ARGs or mobile genetic elements (MGEs) detected in samples. Proximity to Antarctic stations was correlated with detection of ARGs. *blaTEM*, *blaSFO*, *blaFOX*, *cphA*, *mexF*, *oprD*, *oprJ* genes were detected at most sites. Magnesium oxide, pH, and total organic carbon had correlations with ARG distribution.	[17]
Antarctica	Animal feces, fish tissue, seawater	ARB (coliforms)	Isolates from sewage water had more resistance than isolates from pristine areas.	[18]
Antarctica	Marine water, soil, fauna	ARG (Int1 in *Escherichia coli*)	20.7% isolates positive for Int1 from seawater, sediment, and *Laternula elliptica.*	[19]
Antarctica	Soil, rock	ARGs	Tetracycline, betalactamase, vancomycin, and transporter genes from soil and rock microbiomes identified from maritime sites to “extreme inland” sites. Strongest ARG signal in: Halobacteria, Proteobacteria, and photosynthetic bacteria.	[20]
Antarctica	Water, animal feces	ARB	ARBs were proportionally higher in non-native bacteria and higher overall, closer to human activity.	[21]
Antarctica	Soil	ARGs	177 ARGs identified. Vertical inheritance suggested over HGT.	[15]
Antarctica	Soil	ARGs	79 ARGs detected, *bacA* the most common. Four major mechanisms: efflux pumps, bypass mechanisms, target modification, target inactivation. 10 ARGs found in more than 70% of samples.	[22]
Canadian Arctic	Water, ice	ARB (coliforms)	Water isolates demonstrated resistance more than glacial ice isolates. Cefazolin resistance most common (84%). No isolates resistant to streptomycin, gentamicin, chloramphenicol, or ciprofloxacin.	[23]
Swedish Arctic (Abisko, Sweden)	Soil	ARGs	No significant difference in ARG abundances across three permafrost types. Efflux pumps conferring multiple resistances most common ARG type.	[24]
Kongsfjorden, Svalbard	Soil	ARGs	67% of ARG variation due to lithological and nutrient factors. 131 ARGs detected from 9 major classes. MGEs significantly associated with ARG abundances.	[25]
Kongsfjorden, Svalbard	Water, sediment	ARB	High percentage of ARB in sediment and water overall. Percentage of ARB highest in sediment.	[26]
Bering Sea, Polar Research Institute	sediment	ARGs	Sulfonamide genes most prevalent of ARGs tested. All 6 *Tet* genes investigated were found. Human mitochondrial marker (*Hmt*) promising metric for human presence.	[14]
Brazilian national park	water	ARB	93% of isolates resistant to at least one antibiotic at “some level”. Isolates from *Kluyvera* genera were always susceptible. Isolates from rainy season samples had multiple resistances more frequently. Biotic factors were not associated with antimicrobial resistance patterns.	[27]
Swiss lakes	Water	ARGs	Sulfonamide genes most common. *qnrA* not detected in any samples. *sul1* abundance best explained by presence of wastewater treatment plants and hospital effluent. *sul2* best explained by total phosphorous, lake retention time, and urban proximity.	[28]
Argentinian lakes	Water	ARB	All isolates resistant to at least 8/11 tested antibiotics. Every isolate resistant to ampicillin and all macrolides. Arsenite resistance detected in 8/13 isolates.	[29]
South African hot springs	Water	ARB	37.5% of isolates had multiple resistances. 52.5% of isolates resistant to cefepime. No association between heavy metals and ARB.	[30]
Hong Kong marine reserve	Water	ARB (Vibrio)	All isolates demonstrated multiple resistance.	[31]
Creeks in Georgia, USA	Water, sediment, oysters	ARGs	All samples positive for either a *tet* or *int* gene. *tetM*, *tetQ*, and *int1* most commonly detected genes. Significant associations between ARG frequencies and salinity and conductivity.	[32]
Isolated cave system, United States	Water	ARB	Of Gram-positive strains, 70% resistant to 3 or 4 antibiotic classes. Of Gram-negative strains, 65% resistant to 3 or 4 antibiotic classes.	[33]
Poudre River, Colorado, United States	Sediment	ARGs	ARGs found more frequently at impacted downstream sites. ARGs detected at all sites.	[34]
Ningbo, Zhejiang Province, China	Soil	ARGs	212 ARGs, 8 transposons, 1 *int* gene, 1 integron-integrase gene detected. ARGs found significantly more often in farmland than forest.	[35]
Deep ocean and highly impacted estuary, China	Sediment	ARGs/Abx	ARG abundance trended with antibiotic concentrations for each sample. *macB* and *acrB* most commonly detected ARGs in deep ocean sediments.	[36]
Tibetan lakes	Sediment, soil, feces	ARGs/Abx	No sulfonamide, tetracyclines, fluoroquinolones, or macrolide antibiotics found in any samples. ARGs more abundant in sediment than soil or animal waste.	[37]
Glaciers around the world	Snow/ice	ARGs	48.4% of tested ARGs found in snow/ice samples. 3.2% of tested ARGs found in ice cores. *Aac3* most commonly detected ARG (44.4%). Central Asian and Himalayan glaciers had highest relative ARG abundances.	[38]

**Table 2 ijerph-17-03939-t002:** Pros and cons of common ARB/ARG analytical methods.

Method	Pros	Cons
Traditional Agar Plate	1. Can be used for general bacteria or indicators	1. Expensive per sample
	2. Not technically difficult and highly standardized	2. Optimization required for every sample to achieve countable plates
	3. Confirms bacterial viability for risk determination	3. Time and effort requirements limit sample sizes
		4. Extreme limitation of culturable environmental species
Flow Cytometry	1. High-throughput. Can incorporate large sample sizes.	1. No standard methods for investigating antibiotic susceptibility
	2. Can be used to assess bacterial communities as well as single taxonomic groups	2. Expensive up-front costs
	3. Reliable absolute viable and nonviable bacterial concentrations with low limit of detection	3. Requires an instrument capable of resolution to discriminate bacteria
		4. Frequent equipment calibration
Targeted Genomic Analysis	1. High-throughput. Can incorporate large sample sizes.	1. Does not assess bacterial viability
	2. Low limit of detection for targeted ARGs or taxonomic groups	2. Not a good screening tool for ARGs without a priori knowledge
	3. Relatively low cost compared to traditional agar plate method	3. Abundances are relative to standard curves
Metagenomic Analysis	1. A good screening tool for assessing ARGs present in a community	1. Does not assess bacterial viability
	2. Can develop phylogenetic trees for microbial communities in samples	2. Requires extensive technical knowledge and computing power to analyze data
	3. High-throughput. Can incorporate large sample sizes	3. Analysis of assemblies requires public databases and are subject to their biases
		4. Assemblies rely on present bacterial community, minimizing comparability between samples
		5. Cannot determine absolute abundances of organisms or genes

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
