# Peer review of "Antibiotic Resistance in Minimally Human-Impacted Environments"

_ijerph, 2020, doi:10.3390/ijerph17113939_

Round 1
Reviewer 1 Report
Please see file

Author Response
Reviewer 1
Introduction: I think what is missing here is a paragraph or two outlining why there are ARBs in non-impacted environments to help a non-expert reader to make sense of the data presented in this review. This is briefly mentioned later in the paper, but I think it would be more helpful in the introduction. In addition, it would be helpful to have a paragraph outlining hypothesis regarding the level of resistance observed in for example aquatic vs. soil environments.
Answer: Inserted an introductory paragraph addressing background antibiotic resistance and hypotheses about antibiotic resistance in minimally human impacted environments. L. 56-64.
- 26: Please give examples for biological processes.
Answer: Rewrote sentence to “The antibiotic resistance genes (ARGs) within ARB can proliferate through environments due to persistent human activity and the perpetuation of the genes by horizontal gene transfer and vertical inheritance”. L. 26
L.39f.: “...contamination ... as contaminants”. This sentence is unclear.
Answer: Rewrote sentence to “Furthermore, the study of minimally human impacted environments may help elucidate the extent of contamination of ARB and ARGs from the anthropogenic sources. L. 39-41.
- 42: How do you define “environmental health”?
Answer: Deleted previous sentence and reworded for better clarification. “The issue of environmental antibiotic resistance can be addressed within the One Health paradigm. In order to estimate potential human health risks, the effects of environmental ARGs and ARB on wildlife and the surrounding ecological systems must be understood and quantified.”L. 43-51.
- 47f.: “relatively new” is very vague and I’m not sure what the authors are trying to say as we have studied environmental resistance and resistant isolates from the environment for decades. I would agree that ‘omics’ methods are newer and that most studies are concerning impacted environments (but even then, there are papers more than a decade old that look at ancient resistance in permafrost cores).
Answer: This sentence was rewritten to correctly reflect the span of time environmental antibiotic resistance has been studied in relation to public health. “The focus of environmental ARGs and ARB as a One Health issue has been a research priority since 2004”. L64-65.
Ref: Destoumieux-Garzón, D.; Mavingui, P.; Boetsch, G.; Boissier, J.; Darriet, F.; Duboz, P.; Fritsch, C.; Giraudoux, P.; Le Roux, F.; Morand, S., et al. The One Health Concept: 10 Years Old and a Long Road Ahead. Front Vet Sci 2018, 5, 14-14, doi:10.3389/fvets.2018.00014.
- 95ff.: Please explain why this is interesting.
Answer: Expanded the sentence to include contrary results from an urban river in India determining water parameters are significantly associated with antibiotic resistance patterns. L. 123-125
Section 3.2.: There is a lot more statistical detail given in this paragraph compared to the previous one. This reads a bit strange (as if two different people have written the manuscript) and I recommend to give the same (high or low) level of detail throughout the whole article.
Answer: Statistical detail was eliminated from this paragraph to be comparable with other paragraphs.
Table 1: I really like this table, but I was wondering how studies were chosen to be included in it? Did the authors do a literature search suing specific key words, for example? The table header could be more informative.
Answer: The Table 1 header was expanded to include search engines and terms used. Additionally, these criteria were outlined in the introduction (L. 74-78).
- 138-149: Interesting, but a counter-argument would be that due to the plate count anomaly it is impossible to find a ‘generalized’ medium.
Answer: Inserted a statement about the bias of generalized media. “However, the utilization of generalized media to determine antibiotic susceptibility among a larger variety of bacteria taxa will result in a competitive bias towards fast-growing groups.” L. 193-195.
- 150-159: Human-associated, well-described indicator species will also be extremely rare or absent in most remote environments and might have been much more likely introduced during the sampling.
Answer: Inserted a statement reflecting the bias of indicator species and human impact “However, human-associated, well-described indicator species will also be extremely rare or absent in most environments with minimal human activity and this could introduce sampling bias. L. 202-204.
- 162ff.: This is very interesting. I would encourage the authors to explain more about how this method works as an introductory sentence. How could different taxa or strains be differentiated? In general, this paragraph is very detailed compared to the previous ones. Is this level of statistical detail necessary?
Answer: Inserted a sentence describing how fluorescent dyes and flow cytometry can be used to determine bacterial viability “The combination of fluorescent cell permeable and impermeable nucleic acid probes and light scattering properties of bacterial particles allows for the viability assessment of bacterial communities”. L. 211-213. Eliminated extraneous statistical detail. Inserted sentence outlining the current inability of flow cytometry to differentiate specific bacterial taxa “Speciation of resistant or susceptible bacteria might not be possible with current flow cytometry technology.” L. 223-224.
Table 2: Only a small fraction of all bacteria is currently culturable. I would definitely list this as a ‘Con’ for the ‘Traditional Agar Plate’. For ‘Flow Cytometry’ I would in addition list the constant need for calibration of flow cytometers (which is methodologically not trivial) as a further ‘Con’. For ‘metagenomic analysis’ further ‘Cons’ are that one does not obtain absolute abundance of ARGs or ARBs and numbers are thus difficult to standardize across samples; the likelihood of detecting ARGs relies on the quality of the assembly, which can be highly community dependent; and homologies between genes may make it difficult to distinguish true resistance genes from non-functional or other closely related genes.
Answer: Additional cons for “Traditional Agar Plate”, “Flow Cytometry”, and “Metagenomic Analysis” were added to Table 2.
- 210ff.: From the authors’ review of the literature, could they analyse which environmental factors (e.g. pH or temperature) may be driving ARG abundances in relatively pristine environments?
Answer: We believe this has been understudied, and given the minimal amount of data about these parameters and their influence on ARG or ARB abundances it is difficult to characterize those relationships.
- 226-232: I see the authors’ point, but I don’t agree completely. I think adding more samples may also confound results, as long as we don’t know what environmental factors may be driving ARG abundance in environments. E.g. to a researcher it might look like adding another soil sample to their study, but from the bacterial communities’ point of view this might be a vastly different environment.
Answer: Included a statement reflecting this counter point and the opportunity it presents to further investigate changes in environmental factors and antibiotic resistance patterns. “However, the inclusion of more samples may introduce the possibility of varying microenvironments and the bias associated with the differing bacterial communities within them. Nevertheless, further investigation of these changes and their effect on the bacterial community while holding constant anthropogenic impact offer the opportunity to characterize these potential drivers of resistance.” L. 286-290.
- 235f.: This sentence is a bit confusing. I don’t think ‘absence’ can be gradual so maybe ‘anthropogenic impact’ would be a better term.
Answer: Changed “anthropogenic absence” to “anthropogenic impact”. L. 294.
Reviewer 2 Report
This review is about antimicrobial resistance in minimally human impacted. There is an urgent need to elucidate what factors are involved in the emergence and dissemination of ARB and ARGs and little is known about minimally human impacted environments, particularly to determine background or intrinsic level. Moreover the authors underline the lack of standardized methods that makes difficult the quantification of environmental resistance, and so the comparison of antimicrobial resistance from diverse environments.
Minor revisions:
Genes have to be written in italic (L84, L100, L115, 116, table 1)
L105: as its point is surprising, can you briefly explain the link between spore forming ability and antimicrobial resistance?
Table 1 is not referenced/notified in the text.
Table 1: instead of CEF, can the entire name of the molecule (Cefepime) be written
Author Response
Reviewer 2:
Genes have to be written in italic (L84, L100, L115, 116, table 1)
Answer: Genes have been italicized
L105: as its point is surprising, can you briefly explain the link between spore forming ability and antimicrobial resistance?
Answer: The mechanism between spore forming and increased antibiotic resistance remains unclear. The sentence was rewritten to reflect only the correlation between spore forming and increased antibiotic resistance in this paper and previous studies.
Table 1 is not referenced/notified in the text.
Answer: Inserted reference to Table 1 (L. 74-78).
Table 1: instead of CEF, can the entire name of the molecule (Cefepime) be written
Answer: “CEF” changed to “Cefepime” in Table 1.